# Value-Based Care in Systemic Therapy: The Way Forward

**Aju Mathew [1],\*, Steve Joseph Benny [2], Jeffrey Mathew Boby [3] and Bhawna Sirohi [4]** 

1   Department of Oncology, MOSC Medical College, Ernakulam 682311, Kerala, India
2   Government Medical College, Thrissur 680596, Kerala, India
3   Government Medical College, Kozhikode 673008, Kerala, India
4   Department of Oncology, Balco Medical Center, Raipur 493661, Chattisgarh, India
\*   Correspondence: cancerkerala@gmail.com

**Abstract:** The rising cost of cancer care has shed light on an important aspect of healthcare delivery. Financial toxicity of therapy must be considered in clinical practice and policy-making. One way to mitigate the impact of financial toxicity of cancer care is by focusing on an approach of healthcare delivery that aims to deliver value to the patient. Should value of therapy be one of the most important determinants of cancer care? If so, how do we measure it? How can we implement it in routine clinical practice? In this viewpoint, we discuss value-based care in systemic therapy in oncology. Strategies to improve the quality of care by incorporating value-based approaches are discussed: use of composite tools to assess the value of drugs, alternative dosing strategies, and the use of Health Technology Assessment in regulatory procedures. We propose that there must be a greater emphasis on value of therapy in determining its use and its cost.

**Keywords:** value; value-based; systemic therapy; cancer; financial toxicity; cost of care

## 1. Rising Cost of Cancer Care

Cancer has emerged as one of the leading causes of mortality and morbidity worldwide with nearly 19.3 million new cases and 9.9 million deaths being reported across the globe in the year 2020. Adding on to the rising prevalence and debilitating disease course associated with cancers, in recent years there has been a tremendous rise in the cost of cancer care. The national cost of cancer care in the USA in the year 2015 was around $183 billion and it is estimated to grow up to $246 billion by 2030 [1]. Financial toxicity has been widely accepted as a serious adverse effect of cancer treatment with deleterious consequences to the patient and family. It is broadly defined as the adverse financial situation that arises as part of cancer therapy for the patient, and it directly impacts their financial security and well-being. The rising economic burden of cancer therapies is associated with an increased risk of financial insolvency and non-compliance to treatment in addition to negatively impacting the quality of life of the patient and family [2]. A study conducted among cancer patients in the USA reported that patients with cancer are 2.6 times more likely to go bankrupt compared to those without it [3]. The rising cost of cancer care is especially prominent in low-middle-income countries with low government expenditure on health care, dismal insurance coverage, and high out-of-pocket expenditures [2].

## 2. Burgeoning Drug Prices

Although the cost of cancer therapies depends on a multitude of factors, drug prices have played a major role in its rise in recent years. Novel cancer drugs with minimal or no improvement in clinical outcomes are often marketed at much higher prices compared to their predecessors. Global spending on anticancer drugs crossed $150 billion in 2018 and is expected to exceed $240 billion by 2023 [4]. In the USA, median launch prices of novel anti-cancer drugs rose by around 100 times in the last six decades, with the prices rising from $100 a month in 1960 to nearly $10,000 in 2014 [5]. Adding on this, the monthly cost

of newer drugs rose by 5% every year after adjusting for inflation with an extra 10% for every additional indication approved by the FDA [6]. The situation is much more grave in LMIC that have limited access to even essential anticancer drugs [7].

## 3. Cost versus Benefit

Contrary to common belief, more expensive drugs are not always associated with greater clinical efficacy or better survival. A study analysing the clinical utility of newly approved anticancer drugs in Europe reported that significant improvements in survival and quality of life were seen with only 35% and 10% of the newly approved drugs respectively [8]. Another study evaluating 71 consecutive chemotherapeutic agents licensed between 2002 and 2012 reported only a minimal increase in median overall survival and progression-free survival of 2.1 and 2.3 months respectively [9]. Apart from the poor cost-effectiveness, approval of low-quality interventions would also hinder progress in the field of cancer care. Fojo et al. argue that approving therapies with dismal clinical improvements would decrease the incentive for the development of high-value drugs and in turn, hamper the development of high-quality clinical intervention [9].

## 4. A Step to Mitigate Financial Toxicity

The rising incidence of cancer coupled with a high prevalence of financial toxicity among cancer patients calls for immediate measures to mitigate the economic burden of cancer therapies. Value-based care is one such intervention. This is a model of care where high-quality interventions with increased efficacy, improved clinical outcomes, and lower cost are encouraged, and there is limited spending on low-value interventions that are more expensive and associated with poorer health outcomes.

The essence of the term "value" has not only been reviewed by medical professionals, but also by economists, all of them stating their perspectives concerning their respective fields. Economist Adam Smith brought the concept of the diamond–water paradox in his book 'The Wealth of Nations' wherein he compared the low price of water, without which even life is impossible (therefore, invaluable), to the high price of a diamond, which is not an essential factor for life [10]. Hence, a commodity's price does not always reflect its worth.

## 5. Value in Healthcare

The quantified progress in a patient's health outcomes for the expense of attaining that progress is referred to as value in healthcare [11]. The concept of value-based healthcare challenges many of the currently employed clinical practices, mainly curative and preventive interventions that have poor outcomes despite their high cost [12]. The ultimate goal of value-based healthcare is to provide better and fair standards of healthcare services with optimized utilization of healthcare resources [13]. Hence depictions of value that focus on cost reduction alone misinterpret the actual notion of value-based medicine.

Adequate and timely management of a disease lowers the chances of associated complications and worsening of the disease, and thereby cuts down on treatment expenses for the patient in the long run. This approach is particularly relevant in the treatment of chronic diseases. For instance, a well-managed patient with diabetes who does not progress to nephropathy, blindness, or neuropathy, is saved from additional expenses that they would have had to bear if the condition was not well controlled [14].

Many studies have brought forth schemes to implement value-based healthcare. Teisberg and Wallace devised a framework for transitioning from volume-based to value-based healthcare [15]. It includes understanding the shared health needs of patients, designing comprehensive solutions to improve health outcomes, integrating learning teams, measuring health outcomes and costs, and expanding partnerships. Providing patients with an all-inclusive solution to their problems that addresses issues causing poor patient compliance has been observed to produce excellent health outcomes. Porter and Teisberg have proposed a similar concept, which they refer to as an integrated practice unit (IPU) that

comprises a dedicated team that caters to both the medical and non-medical needs of a patient, offering a comprehensive solution to the patient's condition [12,16]. For example, a patient with migraine headaches would also be provided with psychological counselling, physical therapy, and relaxation training in addition to medications.

In any case, value-based care is of paramount importance to LMICs like India. Financial toxicity in cancer care is rampant among LMICs. In India, almost 60% of all hospitalization in rural areas and 40% in urban areas are met through distress financing (i.e., by the selling of assets, pawning of jewellery, borrowing money, or using up of all savings) [17].

## 6. Measuring Value: The ASCO and the ESMO Tools

Acknowledging the importance of the economic burden of cancer care, the American Society of Clinical Oncology (ASCO) developed a value framework to enable physicians to systematically assess the value of newer cancer drugs [18,19]. The European Society for Medical Oncology (ESMO) also developed a similar tool, the Magnitude of Clinical Benefit Scale (MCBS) to help clinicians choose high-quality interventions [20,21]. The salient features of the two scales are summarised in Table 1.

**Table 1.** Tools to assess value of systemic therapy in oncology.

| ASCO Value Framework | ESMO MCBS |
| --- | --- |
| <ul><li>Enable physicians to systematically compare the value of new drug or intervention against existing standards of care</li><li>2 separate versions have been devised one for potentially curative cancer and the other for advanced cancer</li><li>Here clinical efficacy and toxic profile of the new drug is analyzed to determine Clinical Benefit and Toxicity Score</li><li>Clinical Benefit Score can be calculated using Hazard Ratio, Overall survival, Progression-free survival, etc</li><li>Clinical Benefit Score and Toxicity Score are combined to give Net Health Benefit Score which is then compared against the cost of treatment</li><li>In advanced cancer framework, a bonus point is awarded for improved palliation, quality of life, and treatment-free interval compared to standard treatment</li></ul> | <ul><li>To facilitate physicians in determining the value of anti-cancer therapies</li><li>3 separate versions have been devised for potentially curative therapies, therapies not likely to be curative, and therapies for "orphan diseases" or diseases with "high unmet needs"</li><li>In curative settings, the therapies are classified into A, B, and C grades with A being the highest grade. Grade A and B therapies are associated with substantial clinical benefits.</li><li>In a non-curative setting, therapies are classified into 1,2,3,4, and 5 grades with 5 being the highest and 1 being the lowest grade. Grades 4 and 5 are associated with substantial clinical benefits.</li><li>The grades are determined based on pre-set criteria that depends on toxic effects, overall survival, disease-free survival, hazard ratio, progression-free survival, quality of life, etc of the therapy being assessed</li></ul> |

Although there are frameworks evaluating the value of clinical interventions, it is not without limitations. The estimation of value based on the ASCO Value framework is complex and involves elaborate calculations, thus limiting its regular use in routine clinical practice. The complicated process involved in the determination of the Net Health Benefit Score resulted in the same regimen having different values among different clinicians, thus lowering the inter-reliability of the ASCO Value framework [22]. The Clinical Benefit Score when calculated using HR (hazard ratio) is different from that calculated with the Overall Response Rate (ORR) for the same regimen. Heterogeneity in clinical benefit scores based on the variable used in its calculation is another demerit of this value framework [23]. Both the ASCO Value framework and ESMO MCBS score depend on the control arm used for calculation. Hence the value of a particular drug can vary depending on the efficacy of the control arm.

## 7. Role of Regulatory Agencies

Many of the recent cancer drugs are approved based on surrogate endpoints, such as progression-free survival (PFS) or response rate (RR) [24]. Patients benefit from such endpoints in that they reduce the delay in newer drugs entering the market [25]. However,

these drug approvals are not based on endpoints, such as overall survival (OS) and health-related quality of life (HRQoL), which are widely considered as the primary objectives for the use of a medication – it should help a person live longer or live better [25]. This can result in high-priced drugs entering the market that do not improve OS or be harmful to the patient [24]. It is therefore essential to conduct phase III trials and post-marketing studies of drugs that have been approved on the basis of surrogate endpoints to establish their impact on overall survival [25].

In contrast, the clinical benefits of many of the drugs that were approved through the accelerated approval pathway are not being confirmed by phase III randomised control trials [25,26]. Confirmatory trials, if performed, are often unable to establish the clinical benefits of many such drugs. Hence, many of the drugs that are being marketed are not found to have an impact on OS or HRQoL [8]. Regulatory agencies must set the bar higher so that clinically meaningful improvement in therapy can be achieved with the approval of a new therapy. In other words, drug approvals on the basis of surrogate endpoints and the lack of post-approval confirmatory trials raises the possibility for the use of low value drugs in oncology.

## 8. Implementing Value-Based Care in Oncology: Strategies for the Clinic

Increased cost communication between patients and oncologists can also help to improve the value of care. Even in high-income countries, although a large proportion of patients are interested in such discussions, it seldom happens in real world clinical practice [27]. Such discussions may enable the patient to make well-informed decisions on their treatment. This may help in better management of assets, increased treatment adherence, and improved clinical outcome. Discussion on the cost of cancer therapies early in the course of the disease can help in the timely identification of patients affected by financial toxicity [2]. The use of the COST (Comprehensive Score for financial Toxicity) questionnaire can help in this regard, though this is not validated for LMICs [28]. Such patients can then be offered the services of financial navigators, counsellors, or welfare workers.

Early incorporation of palliative care can also improve the value of cancer care delivery. Integration of palliative care into patient management is associated with decreased number of emergency room visits and inpatient hospitalization, reducing the cost of care in addition to improving patients' quality of life [29,30]. Patient Reported Outcome Measures (PROM) is another novel intervention that can enable the delivery of high-quality care. PROM is a well-validated tool that helps in understanding the symptoms of disease, adverse events associated with treatment, quality of care received, and response to therapeutics from a patient viewpoint [31]. It helps in assessing the patient's thoughts and opinions on the care received. In addition to enhanced patient satisfaction and better patient–doctor relationship, regular use of PROM may even have a survival benefit [32].

Apart from encouraging high valued clinical practices, measures should also be taken to limit the delivery of low-quality care. It has been estimated that around $75.7 billion to $101.2 billion is lost every year in the US as a result of low-value care and overtreatment [33]. Clinical pathways are another tool that can be utilized to ensure high-quality cancer care. Adhering to validated clinical pathways ensures that high valued treatment guidelines are followed, and evidence-based medicine is practiced [34]. Diagnostic and laboratory tests should be ordered only if the results have an impact on further treatment strategies [35]. Moreover, these pathways must be designed in a patient-centered manner that takes into account both the patients' and physicians' perspectives in decision making.

## 9. Alternate Dosing Strategies

Many anticancer drugs are often administered at doses much higher than what is actually required. Ibrutinib, which acts on Bruton Tyrosine Kinase (BTK), saturates the BTK site at a dose of 175 mg/day for a 70 kg individual, but it is marketed at a dose of 420 to

560 mg per day [36]. Hence cutting the administered dose by half would possibly decrease the cost of treatment by 50% while providing the same clinical benefit.

The pharmacokinetic properties of drugs can also be exploited to lower the dose of cancer agents. The dose of abiraterone, a drug used in prostate cancer, can be reduced by 5 to 10 times if taken with food even though it is labelled to be administered on an empty stomach [37]. Hence if we were to exploit the food effects of abiraterone, it would lead to tremendous savings in the long run. Cutting down on doses of anticancer drugs to what is adequate is often much easier and more practical than policy reforms to decrease the per mg cost of these agents. In addition, it also helps to decrease the magnitude and frequency of toxic effects associated with these anticancer therapies [36].

Immune checkpoint inhibitors (ICI) have a distinct mechanism of action when compared to other oncologic drugs [38]. Their pharmacodynamics differ notably among patients receiving the treatment [38]. The anti-neoplastic effect of ICIs is observed to not be dependent on the amount of the drug administered [39]. Studies have shown that the objective response rate (ORR) of pembrolizumab, an ICI, remained stable despite changing its dosage [40]. The high level of pharmacodynamic variability of ICIs necessitates the development of an effective dosing strategy that would deliver the best possible clinical outcomes to patients at reasonable rates.

Bodyweight-based dosing regimens were proposed for many of the ICIs that assumes a direct relationship between drug clearance and body weight [41]. However body weight-based dosing does not ensure uniform drug exposure to the patients. Moreover, it has led to increasing concerns about drug wastage from discarded single-dose vial packages [42]. The limited availability of vial sizes for drugs administered in weight-based dosing leads to a fraction of the drug being left unused in vials that are eventually discarded as they do not always match up accordingly with each patient's body weight.

Studies also evaluated the benefits of the transition to flat dosing over weight-based dosing. It was found that apart from minimizing drug wastage, flat dosing also saves time from prescription to production [38]. The problem encountered in fixed dosing strategies is that they can result in inadequate exposure in patients with high body weight and overexposure in patients with low body weight [43]. In addition to this, a study by Goldstein et al. comparing the annual cost of pembrolizumab therapy based on fixed dosing and personalized dosing claimed that the latter regimen was much more cost-effective, leading to an annual saving of more than $800 million for non-small-cell lung cancer in the US alone [44]. The aftermath of such a treatment modality is that it results in the wastage of resources on drugs [45].

## 10. Pricing Drugs Proportional to Clinical Benefit

Another concern regarding treatment expenditure in cancer care is that the cost of drugs is not determined by their clinical benefits. The majority of anticancer drugs are a part of multiple treatment regimens with varying clinical benefits [46]. However, despite the uncertainty in patient outcomes in many cases, the price of these drugs remains the same. For instance, the drug pertuzumab is observed to have a low value in adjuvant therapy compared to its high value in the metastatic setting, but the cost of the drug remains the same in both scenarios. Hence, irrational pricing of drugs is a hindrance to achieving value-based care in oncology. The clinical benefit of a drug could be assessed separately for each of its indications, and the prices set accordingly against the value, as determined by an accepted scale [46]. Health technology assessment (HTA) is a form of policy research that aids in providing information regarding diverse healthcare aspects [47]. A lack of expertise and inadequate government resources are barriers to its implementation in LMICs.

Although value-based care is being increasingly adopted across the globe, the strategies employed, the source of funding, and the major players involved significantly vary among different countries. The Economist Intelligence Unit assessed the health systems in 25 countries, divided nations based on the degree of alliance towards value-based health care. Sweden and UK were the only nations reported to have "very high alliance" and "high

alliance" respectively [48]. Although many nations collect data on patient treatment costs, it is limited to specific geographical locations and a comprehensive system exist only for nations like Sweden, South Korea and Germany [48]. Independent organizations for health technology assessment, though present in developed nations like Germany, Netherlands and Sweden, are absent in the US and Japan [48]. Countries like France, Sweden and UK have systems in place that identify and de-adopt intervention with poor value, but these mechanisms are also lacking in Japan and the US [48]. Hence even though the goals of value-based care remain largely the same, the measures employed in pursuit of it shows characteristic difference among nations.

## 11. Conclusions

It is essential to realize that value-based health care is ultimately not just a solution to the problem of rising drug prices. It is the first step in aligning the best interest of patients with clinical care. What matters to the patient must be what is valuable for the healthcare setting (living longer with a good quality of life). Value-based care in cancer therapy must be a means to that end.

**Author Contributions:** Conceptualization, A.M., S.J.B., J.M.B., B.S.; writing—original draft preparation, A.M., S.J.B., J.M.B.; writing—review and editing, A.M., S.J.B., J.M.B., B.S.; supervision, A.M.; project administration, A.M. All authors have read and agreed to the published version of the manuscript.

**Funding:** This research received no external funding.

**Conflicts of Interest:** The authors declare no conflict of interest.

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
