# Peer review of "Value-Based Care in Systemic Therapy: The Way Forward"

_curroncol, doi:10.3390/curroncol29080456_

Round 1

Reviewer 1 Report

I recognize the importance of carrying out this type of literature review to operate on certain variables that may have an impact on a specific problem, as well as how to approach it. In this sense, the objective of this review is to address the economic impact that cancer treatment has on the health of patients and their families.

The first criticism I have about this study is that I would recommend the authors to carry out a meta-analysis or at least a systematic review. In fact, this would allow checking if the authors' interpretation were biased (e.g., selecting the literature that only reinforces their starting hypothesis, etc.). In fact, to give transparency to the process, it would be necessary to add information on the methodological process to select the literature and if an article was left out of the review.

The authors ignore the countries in which health is public and, therefore, the cost of the medication should not be borne by the patient himself. This would be a good "control group". However, they do not refer to these countries.

The authors include some qualitative information on the opinion of certain oncologists. In fact, section "6. A Survey of Oncologists on Value-based Care" seems fundamental to the study. Even so, the authors do not refer to this information in the abstract. Therefore, it would be necessary to restructure the article, since if the article is a review, it is very strange to include empirical information that has not been published.

It is necessary to completely restructure the article to be a review of the literature. In fact, as it is written, it looks like an opinion piece or the presentation of the design of a project. Therefore, the first thing I suggest is to write an abstract with the following sections: introduction, methodology, results, and conclusions. In fact, it is necessary to clarify the objective of the study in the abstract and in the introduction.

It would be necessary for the authors to refer to the articles that have addressed the impact of the problem they describe on the health of patients. Similarly, certain statistics could be introduced to allow results to be compared between countries with different health systems.

Finally, it would be necessary to reinforce the conclusions of the study, since the authors have been very concise compared to the rest of the article.

Author Response

The first criticism I have about this study is that I would recommend the authors to carry out a meta-analysis or at least a systematic review. In fact, this would allow checking if the authors' interpretation were biased (e.g., selecting the literature that only reinforces their starting hypothesis, etc.). In fact, to give transparency to the process, it would be necessary to add information on the methodological process to select the literature and if an article was left out of the review.

It is necessary to completely restructure the article to be a review of the literature. In fact, as it is written, it looks like an opinion piece or the presentation of the design of a project. Therefore, the first thing I suggest is to write an abstract with the following sections: introduction, methodology, results, and conclusions. In fact, it is necessary to clarify the objective of the study in the abstract and in the introduction.

Response: We believe the reviewer has misunderstood. Our work was intended to be an perspective piece on the theme and not a review.

The authors include some qualitative information on the opinion of certain oncologists. In fact, section "A Survey of Oncologists on Value-based Care" seems fundamental to the study. Even so, the authors do not refer to this information in the abstract. Therefore, it would be necessary to restructure the article, since if the article is a review, it is very strange to include empirical information that has not been published.

Response: Thank you for the comment. Our work was intended as a perspective piece. But we accept the critique and hence we have removed the section “A Survey of Oncologist on Value-based Care”

The authors ignore the countries in which health is public and, therefore, the cost of the medication should not be borne by the patient himself. This would be a good "control group". However, they do not refer to these countries.

It would be necessary for the authors to refer to the articles that have addressed the impact of the problem they describe on the health of patients. Similarly, certain statistics could be introduced to allow results to be compared between countries with different health systems. Finally, it would be necessary to reinforce the conclusions of the study, since the authors have been very concise compared to the rest of the article.

Response: There is a paucity of outcome data on the value of value-based healthcare in oncology.

Reviewer 2 Report

General remarks

The manuscript is a review -- and as so that, of course, will be the subject of this review report -- on the use of a value-based approach to health care in systemic therapy in oncology.

Specific remarks

I really enjoyed reading the manuscript, not least because it illustrates very well some very well-known issues in the field of Health Economics, for which the authors, due to their training, may not be so aware. It is these issue, which I will be present below, that I recommend that authors consider in their manuscript.

First, it is often claimed (in Health Economics) that there is a potential conflict between economists and health care professionals. This is because economists generally have (or should have) a general/global view of the benefits and costs (i.e. not just for the patient, in particular) while health professionals take a particular view, i.e. with regard to the patient, before whom no matter what costs are necessary to incur to restore the patient’s state of health. On this subject, it would be very interesting to go through the literature on the, shall we say, economic value of life. See, among others, Broome, J. (1985). The economic value of life. Economica, 52(207), 281-294.

From that conflict, the (very) probable -- using the authors’ term, -- financial toxicity of certain therapies becomes more prevalent when, for example, healthcare is provided in healthcare systems a la Beveridge, but can also happen in healthcare systems a la Bismarck.

Secondly, it is also customary to say (in Health Economics) that, very often, there is an asymmetry in the information available by the demand and supply of health care, so that supply makes decisions (regarding therapy) that 'imposes' to patients, often without taking into account the budgetary constraints that demand (for health care) almost always has.

That is a second possible source of financial problems, which the manuscript so well illustrates. Thus, it is up to the supply of health care, even because it has the technical knowledge, to prescribe the best therapy, i.e. one that is based on its real value. This problem is particularly important in certain diseases, such as those of an oncological nature, but also in many of a neurological nature, where treatments are prohibitively expensive, as the authors are well aware. This will not be a utopian vision if healthcare providers were able to resist the natural tendency on the part of the pharmaceutical industry to sell new, even more expensive therapies.

Finally, I also recommend a better motivation of the manuscript, possibly in section 1. Rising Cost of Cancer Care, through a simple analysis of the global burden of oncological diseases (in India), as reproduced in the figure below (made at https://vizhub.healthdata.org/gbd-results/ on June 29, 2022). (The figure will not be visible 'online', but will be present in the file I will attach.)

Author Response

I really enjoyed reading the manuscript, not least because it illustrates very well some very well-known issues in the field of Health Economics, for which the authors, due to their training, may not be so aware. It is these issue, which I will be present below, that I recommend that authors consider in their manuscript.

First, it is often claimed (in Health Economics) that there is a potential conflict between economists and health care professionals. This is because economists generally have (or should have) a general/global view of the benefits and costs (i.e. not just for the patient, in particular) while health professionals take a particular view, i.e. with regard to the patient, before whom no matter what costs are necessary to incur to restore the patient’s state of health. On this subject, it would be very interesting to go through the literature on the, shall we say, economic value of life. See, among others, Broome, J. (1985). The economic value of life. Economica, 52(207), 281-294.

From that conflict, the (very) probable -- using the authors’ term, -- financial toxicity of certain therapies becomes more prevalent when, for example, healthcare is provided in healthcare systems a la Beveridge, but can also happen in healthcare systems a la Bismarck.

Secondly, it is also customary to say (in Health Economics) that, very often, there is an asymmetry in the information available by the demand and supply of health care, so that supply makes decisions (regarding therapy) that 'imposes' to patients, often without taking into account the budgetary constraints that demand (for health care) almost always has.

That is a second possible source of financial problems, which the manuscript so well illustrates. Thus, it is up to the supply of health care, even because it has the technical knowledge, to prescribe the best therapy, i.e. one that is based on its real value. This problem is particularly important in certain diseases, such as those of an oncological nature, but also in many of a neurological nature, where treatments are prohibitively expensive, as the authors are well aware. This will not be a utopian vision if healthcare providers were able to resist the natural tendency on the part of the pharmaceutical industry to sell new, even more expensive therapies.

Finally, I also recommend a better motivation of the manuscript, possibly in section 1. Rising Cost of Cancer Care, through a simple analysis of the global burden of oncological diseases (in India), as reproduced in the figure below (made at https://vizhub.healthdata.org/gbd-results/ on June 29, 2022). (The figure will not be visible 'online', but will be present in the file I will attach.)

Response: Thank you for the comments. We think such a discussion is beyond the scope of the manuscript.

Reviewer 3 Report

The paper by Mathew et al. on “Value-based care in systemic therapy: a Utopian vision? is submitted as a review. The topic of value in health care is a huge and important topic even when focused on systemic therapy for cancer. If this was truly a review, there should be a section on the methodology used to select articles for the review. Although many strategies are listed that might help to reduce cost of drugs, it is an incomplete list of approaches.  There does appear to have been a systematic approach to identify relevant article on value  as it relates to systemic therapy.  For example, there is virtually nothing presented on efforts by counties such as the United Kingdom, Canada, Australia, France or Germany to evaluate and approve drugs on the basis of their value for money or in the case of some countries, to involve patients in the assessment of value.  The authors would be better to write a perspective piece from the point of view of a LMIC and describe how the various global efforts to introduce value-based care can best be implemented in LMICs like India.  They could attempt to address the question in their provocative title, is it really a “Utopian Vision” to try to implement strategies to achieve value-based care in LMICs. 

The abstract is really an introductory paragraph and poses the question about the importance of value in the context of cancer care, how to measure it and how it should be implemented in routine clinical practice. If the paper is eventually accepted for publication, the authors should prepare a suitable abstract and use the current abstract as an introduction.

Much of the paper focuses on the cost of cancer care and particularly the high cost of drugs.  The examples given are all from the United States. The reliance on American data with regard to drug costs in an article on value is misguided as there is little to no effort in the US to address value because the practice of medicine is predominantly a business. Being first to implement new practices and to achieve profits are commonly higher priorities despite efforts by ASCO. The article would be stronger if it provided examples of how value in health care is being addressed in various publicly funded healthcare systems around the globe.

In particular, there should be discussion of the use of health technology assessment and cost- effectiveness and cost-utility analyses as one way to try to ascertain the value of a new product.  The authors should recognize that this is not done in the USA as there is specific legislation that precludes Medicare/Medicaid from negotiating price with manufacturers.  There could also be some discussion of the use of real world evidence to determine whether new drugs that are often approved on the basis of immature data are truly delivering value in the real world

Information is provided on the financial toxicity of drugs in the United States but the implications of high drug costs in the developing world are not presented.  It would be informative to the reader to have a better understanding of the limitations in access to drugs considered to be effective and of value to patients in developing countries. What cancer drugs are on the Essential Medicine list and are actua;;ly available in India for example.

The section in the paper describing a survey of Indian oncologists’ knowledge and perspective on value based care is weak. There is no description of how respondents were identified and contacted, what questions were asked and how many responses were obtained etc.

I would suggest that the authors develop a section on strategies that might move health systems towards value based care beginning with the use of measures to assess value (Value Frameworks) , communicating with patients about cost and engaging patients in determining value and then how best to determine the price of drugs.

They should discuss whether these strategies are doable in developing countries or not. As the title of the paper asked the question of whether value based care is in fact a “Uutopian vision”, the authors should attempt to answer their own question as to how realistic it is to implement value based care in LMICs.

The paper requires considerable polishing.  I found it difficult to read as its use of language is awkward at times and words are used in unusual ways (eg. “sanctioned” when speaking of drug approvals).  There is an unnecessary use of colloquialisms such as “a step in the right direction”.

Author Response

The paper by Mathew et al. on “Value-based care in systemic therapy: a Utopian vision? is submitted as a review. The topic of value in health care is a huge and important topic even when focused on systemic therapy for cancer. If this was truly a review, there should be a section on the methodology used to select articles for the review. Although many strategies are listed that might help to reduce cost of drugs, it is an incomplete list of approaches.  There does appear to have been a systematic approach to identify relevant article on value  as it relates to systemic therapy. 

Response: We believe the reviewer has misunderstood. Our work was intended to be an perspective piece on the theme and not a review.

For example, there is virtually nothing presented on efforts by counties such as the United Kingdom, Canada, Australia, France or Germany to evaluate and approve drugs on the basis of their value for money or in the case of some countries, to involve patients in the assessment of value.  

Response: Such a detailed discussion is beyond the scope of this perspective piece. We have added a section where some scattered efforts from some nations are discussed.

The authors would be better to write a perspective piece from the point of view of a LMIC and describe how the various global efforts to introduce value-based care can best be implemented in LMICs like India.  They could attempt to address the question in their provocative title, is it really a “Utopian Vision” to try to implement strategies to achieve value-based care in LMICs. 

Response: In various sections of the article, we have attempted to highlight some ways for improving the quality of care in LMICs – eg; alternative dosing strategies, HTA etc.

The abstract is really an introductory paragraph and poses the question about the importance of value in the context of cancer care, how to measure it and how it should be implemented in routine clinical practice. If the paper is eventually accepted for publication, the authors should prepare a suitable abstract and use the current abstract as an introduction.

Response: We have made changes in the abstract.

Much of the paper focuses on the cost of cancer care and particularly the high cost of drugs.  The examples given are all from the United States. The reliance on American data with regard to drug costs in an article on value is misguided as there is little to no effort in the US to address value because the practice of medicine is predominantly a business. Being first to implement new practices and to achieve profits are commonly higher priorities despite efforts by ASCO. The article would be stronger if it provided examples of how value in health care is being addressed in various publicly funded healthcare systems around the globe.

Response: We have added a small section. But, in our observation, there is paucity of outcomes data from such nations on value-based cancer care.

In particular, there should be discussion of the use of health technology assessment and cost- effectiveness and cost-utility analyses as one way to try to ascertain the value of a new product.  The authors should recognize that this is not done in the USA as there is specific legislation that precludes Medicare/Medicaid from negotiating price with manufacturers.  

Response: We have added a section on HTA.

Information is provided on the financial toxicity of drugs in the United States but the implications of high drug costs in the developing world are not presented.  It would be informative to the reader to have a better understanding of the limitations in access to drugs considered to be effective and of value to patients in developing countries. What cancer drugs are on the Essential Medicine list and are actually available in India for example.

Response: A discussion on the EML is beyond the scope of this manuscript.

The section in the paper describing a survey of Indian oncologists’ knowledge and perspective on value based care is weak. There is no description of how respondents were identified and contacted, what questions were asked and how many responses were obtained etc.

Response: We have deleted the section.

I would suggest that the authors develop a section on strategies that might move health systems towards value based care beginning with the use of measures to assess value (Value Frameworks) , communicating with patients about cost and engaging patients in determining value and then how best to determine the price of drugs.

Response: We have written on tools to assess value.

They should discuss whether these strategies are doable in developing countries or not. As the title of the paper asked the question of whether value based care is in fact a “Uutopian vision”, the authors should attempt to answer their own question as to how realistic it is to implement value based care in LMICs.

Response: We do talk about the various strategies to adapt value-based care in LMICs.

The paper requires considerable polishing.  I found it difficult to read as its use of language is awkward at times and words are used in unusual ways (eg. “sanctioned” when speaking of drug approvals).  There is an unnecessary use of colloquialisms such as “a step in the right direction”.

Response: We have made some changes in the manuscript.

Reviewer 4 Report

The authors do a nice job of discussing value-based approaches as applied to oncology. I suggest a rewording of the title.  I think the point of the original title is to elicit "no" (it is not "just" a utopian vision, which has a negative connotation).  Rather, the authors bring some practical advice about how to achieve such a vision while also pointing out challenges. 

Detailed items: 

Line 25:  it is

Line 30: in the US

Line 41: Adding to this,

Line 44: add acronym LMIC

Line 55:  et al.

Lines 107 and 109: The majority

Line 112: “Patient satisfaction appeared to be a highly valued choice of to attain the goal of value-based care.”  Or “A majority of the participating oncologists appeared to strongly support patient satisfaction as an important component of value-based care.”

Line 119:  the advantages ?  (instead of attributes)

Line 140: practice

Line 188: , and response

Line 260: must be a means

Author Response

The authors do a nice job of discussing value-based approaches as applied to oncology. I suggest a rewording of the title.  I think the point of the original title is to elicit "no" (it is not "just" a utopian vision, which has a negative connotation).  Rather, the authors bring some practical advice about how to achieve such a vision while also pointing out challenges.

Response: We have reworded the title as “Value based care in systemic therapy – the way forward.”

Detailed items:

Line 25:  it is

Line 30: in the US

Line 41: Adding to this,

Line 44: add acronym LMIC

Line 55:  et al.

Lines 107 and 109: The majority

Line 112: “Patient satisfaction appeared to be a highly valued choice of to attain the goal of value-based care.”  Or “A majority of the participating oncologists appeared to strongly support patient satisfaction as an important component of value-based care.”

Line 119:  the advantages ?  (instead of attributes)

Line 140: practice

Line 188: , and response

Line 260: must be a means
Response: We have made all these changes

Round 2

Reviewer 1 Report

---

Author Response

Thank you for reviewing this manuscript.

Reviewer 3 Report

The authors have made a serious effort to address concerns previously raised and have undertaken major revisions.  No additional comments

Author Response

Thank you for the reviewing this manuscript.